# Time pressure alters takeoff but not landing biomechanics in single-leg countermovement jumps

Ugur Yilmaz, Huseyin Celik🆔, Pinar Arpinar-Avsar🆔*

Department of Biomechanics and Motor Control, Faculty of Sport Sciences, Hacettepe University, Ankara, Turkiye

* parpinar@hacettepe.edu.tr

## Abstract

This study examined how time pressure influences lower-limb biomechanics during single-leg maximal countermovement jumps (CMJs), with a focus on kinetic and kinematic responses during both jumping and landing phases. Participants performed single-leg CMJs under two conditions: self-paced (SP) and reaction-time (RT), the latter simulating time-constrained environments. Joint angles, ground reaction forces (vGRF), and joint moments were analyzed. Significant differences emerged between SP and RT tasks in jumping-phase kinetics and kinematics, with only kinematic differences present during landing. The RT condition led to reduced hip and knee flexion, increased peak vGRF, and shorter flight times, yet no improvement in jump height. This suggests inefficient energy transfer possibly due to reduced range of motion and increased muscle co-contraction or pretension strategies. Joint moment analysis revealed a shift from a hip-dominant strategy in SP to a knee-dominant strategy in RT. Landing in RT was characterized by reduced joint flexion and increased frontal plane loading, potentially elevating the risk of lower-limb injury. Time pressure modifies motor strategies in single-leg CMJs, promoting faster execution at the cost of performance efficiency. These findings underscore the importance of training for both explosive performance and neuromuscular control under time-constrained, sport-specific conditions.

## Introduction

Jumping is a key component of performance in field and court sports, with athletes performing between 30 and 120 jumps per match, depending on the sport and playing position [1–6]. Athletes regularly execute both unilateral and bilateral countermovement jumps (CMJs), each involving distinct take-off and landing phases that impose specific biomechanical and physiological demands. Although most research has focused on bilateral CMJs, single-leg take-offs, reported to comprise about 17% of all jumps in basketball, for instance [7], underscore their sport-specific relevance.

**Data availability statement:** All data generated and analyzed during this study are available in the repository at https://doi.org/10.5281/zenodo.17284783.

**Funding:** This research was partially supported by TUBITAK under Grant 115S535 to P.A.A.

**Competing interests:** The authors have declared that no competing interests exist.

Research on single-leg CMJs has primarily explored inter-limb asymmetries [8,9], lower-limb strength and power [8,10], and landing-related injury mechanisms [11–13]. However, studies that simultaneously examine both the kinetic and kinematic aspects of single-leg CMJs remain limited [14]. Such integrated analyses are essential for accurately estimating joint torques and evaluating the explosive capacity of the lower-limb kinetic chain, offering a more comprehensive understanding of segmental contributions, neuromuscular coordination, and power production.

Since the stretch-shortening cycle (SSC) plays a key role in generating explosive power during movements like the countermovement jump [15], optimizing preparatory muscle activity is essential. Recently, it has been shown that generating muscle pretension through co-contraction prior to jump initiation and landing enhances both performance and stability [16]. In the jumping phase, such pretension reduces muscle slack, enabling faster and more efficient force transmission, thereby maximizing the benefits of the SSC [17]. In the landing phase, it also contributes to joint stability and effective impact absorption. Recently, Barrio et al. (2024) suggested that one potential approach to enhance the ability to generate effective pretension is through performing movements under time constraints [16].

Inherently dynamic nature of team sports requires athletes to execute both jumping and landing tasks in constantly changing environments and under varying temporal demands. Time pressure has been shown to affect motor control, with evidence linking temporal constraints to altered movement patterns, increased joint loads, higher take-off velocities, and elevated loading rates [18,19]). To reduce jump duration under time pressure, athletes increase the eccentric rate of force development and concentric peak vertical ground reaction forces [20]. Conversely, landing under unpredictable timing often leads to shorter loading durations and higher impact forces [21], with delayed joint and muscle responses compromising pre-landing stiffness regulation [22].

Although CMJs have been widely studied, many previous investigations lack ecological validity, as they do not incorporate time pressure [23]. To the best of our knowledge, no study to date has specifically examined the impact of time pressure on the three-dimensional mechanical characteristics of single-leg jumping. To address this gap, we analyzed the take-off and landing kinetics and kinematics of single-leg maximal CMJs performed under two conditions: self-paced and reaction-based (in response to an auditory stimulus).

We hypothesized that significant differences would exist in the take-off biomechanics of single-leg CMJs between the reaction-based (RT) and self-paced (SP) conditions, as the reaction-based condition was expected to elicit greater muscle pretension immediately prior to take-off. However, we did not anticipate differences in landing mechanics, assuming that time pressure would primarily influence the jumping phase, and jump height remained consistent across conditions.

## Materials and methods

### Participants

Ten healthy male adults (mean±SD: age, 24.5±3.2 years; weight, 77.5±10.5 kg; height, 177.4±6.9 cm) voluntarily participated in this study. All participants were

at least 18 years of age and classified as recreationally active, engaging in structured physical activity or recreational/competitive sports involving jump-landing movements at least twice per week for a minimum of six months prior to participation.

An a priori power analysis was conducted using G*Power 3.1 to determine the required sample size. Effect size estimates were informed by prior literature: joint angle data at ground contact reported by Edwards et al. [24]; Cohen's $d = 1.16$) and kinetic and kinematic measures from Harry et al. [25]; $d = 1.00$ for kinetics, $d = 0.93$ for joint angles). Based on an anticipated mean effect size of Cohen's $d = 1.03$, a statistical power of 0.80 $(1 - \beta)$, and a significance level of $\alpha = 0.05$, the analysis indicated that a minimum of 10 participants would be sufficient to detect statistically significant effects in a within-subjects design.

All participants were screened for lower-extremity injuries and were free from acute or chronic musculoskeletal conditions (e.g., fractures, ligament ruptures) in the previous 12 months that could influence jump-landing performance. Data collection occurred during the fall semester of the same academic year (between 15/10/2019-15/12/2019). Prior to participation, all individuals were informed of the study procedures and provided written informed consent. The study was approved by the Hacettepe University Research Ethics Committee (Approval No. GO 19/537).

## Testing procedures

The participants performed single-leg CMJs in the biomechanics laboratory using their dominant leg. Force plate and motion capture data were recorded synchronously. Before jumping and landing, the participants were in single-leg quiet stance posture on their dominant leg while hands were on the hip and the non-dominant leg was approximately at 90 degrees knee flexion. The participants performed single-leg maximal CMJ and single-leg landing (with the same leg as in the jumping) task in two different movement-initiating tasks. One of them is the SP task in which after the go signal, in five seconds, the participants should perform a single-leg maximal CMJ whenever they want and to land onto the force plate with the dominant leg. The second one is the RT task in which after the go signal, the participants should perform a single-leg maximal CMJ when they hear a beep signal as auditory stimulus which is randomly generated by the computer within five seconds after the go signal, and to land onto the force plate with the dominant leg. Previous RT paradigm studies generally have focused on auditory and visual stimuli (e.g., [26]). Those studies have shown that response to auditory stimuli is faster than visual stimuli. As we aimed to examine fast movement effects in this study, we preferred auditory stimuli for the RT task.

Each task was repeated five times in block randomization (10 trials in total). To get familiar with the tasks, each participant was allowed to perform each task several times before recording. The participants were asked to rest two minutes between trials to avoid fatigue. Before recording, to identify the dominant leg of the participants, they were asked to kick a soccer ball. Dominant leg was identified as the preferred kicking leg. All participants' dominant leg was their right leg. All trials were performed shoeless.

## Kinetic analysis

The trials were performed on a force plate (AMTI OR6-7-2000, Massachusetts, US). The ground reaction forces and moments were obtained on a three-dimensional global coordinate system which is located on the center of the outer surface of the force plate. The kinetic data was collected at 2000 Hz. To locate the movement initiation frame, the first data point that exceeded 25 N of mean unfiltered vertical ground reaction force (vGRF) during single-leg quiet stance was used (Fig 1) [27]. To locate the take-off frame, the first data point that under 5 N of unfiltered vGRF following the movement initiation frame was used. The movement sequence between movement initiation and take-off was defined as jumping phase (Fig 1). In a similar way, to locate the landing touchdown frame, the first data point that exceeded 25 N of unfiltered vGRF following the take-off was used (Fig 1). The landing phase was defined as the interval between the touchdown and the following 250 milliseconds. The kinetic data were filtered at 10 Hz and interpolated using Woltring's smoothing spline

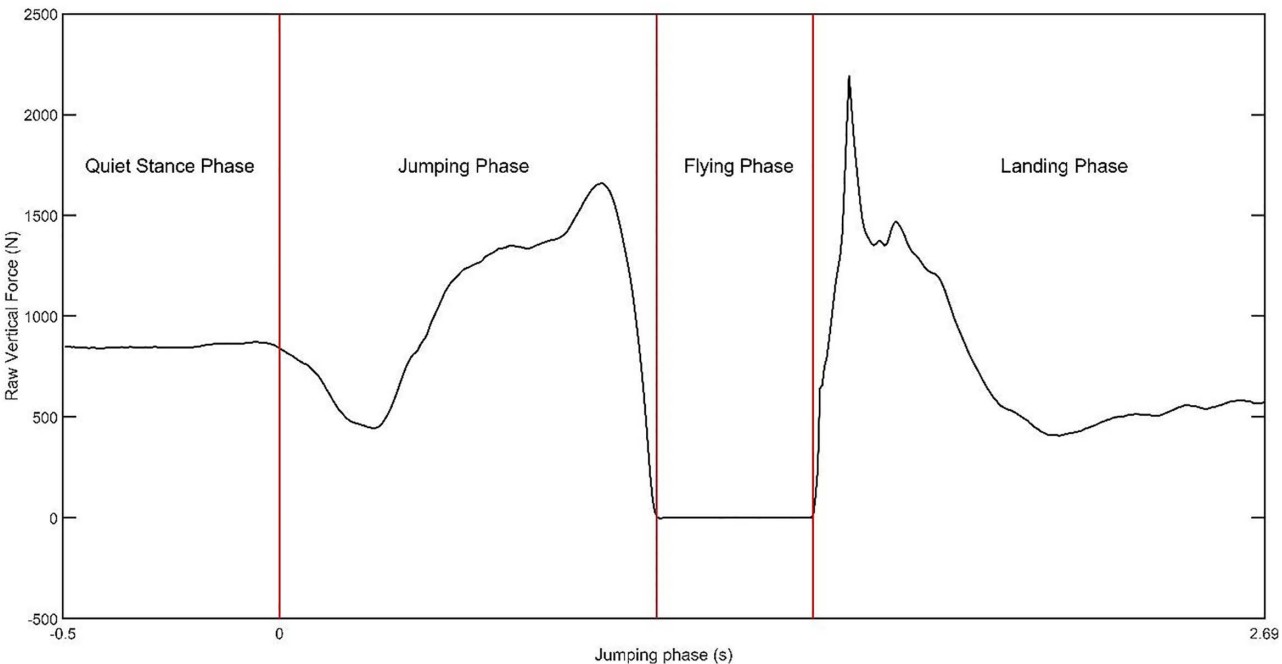

**Fig 1. A typical single-leg maximal CMJ vertical force-time curve indicating the phases of the movement sequence.**

in the cubic mode [28,29]. The filtered kinetic data were used in the inverse dynamics calculations and estimating peak vGRF in jumping and landing phases.

### Kinematics analysis

To estimate joint kinematics, a motion capture system (Optitrack, Natural Point, Oregon, US) with eight infrared cameras (Prime 13) was used for lower body tracking. Specifically, 30 reflective markers (10 mm diameter) were attached to the anatomical landmarks (Rizzoli marker set, [30] to create a biomechanical model which is composed of bilateral thigh, shank and foot, and the pelvis segments in Visual3D software environment (C-motion, Germantown, MD, US). The mass and the moment of inertia of the segments were applied according to Hanavan [31]. The hip joint center was calculated by using a prediction method based on anatomical landmarks of the pelvis [32,33]. The knee and ankle joint centers were defined by calculating the midpoint between medial and lateral femoral epicondyle and the midpoint between medial and lateral malleolus respectively. For pose estimation, built-in 6 degrees-of-freedom algorithm of the Visual3D software was used [34]. The joint angles and moments were resolved in the joint coordinate system (JCS) using a Z-X-Y Cardan rotation sequence. JCS was defined as a right-hand coordinate system, where Z pointed right side (flexion/extension), X pointed anterior (abduction/adduction), and Y was perpendicular to the X-Z plane (internal/external rotation) [35]. The kinematic data were acquired at 200 Hz and also filtered at 10 Hz and interpolated using Woltring's smoothing spline in the cubic mode as the kinetic data [28,29]. The joint moments were normalized by the product of body mass with leg length (the distance from anterior superior iliac spines to medial malleolus) and expressed as the internal joint moment. The jumping height was calculated from the difference of the peak vertical position of the midpoint of right and left posterior superior iliac spines markers between the flight and single-leg quiet stance phases.

The following variables were extracted from the joint angles and moments time series: the peak joint angle and moment values in the sagittal, frontal, and transverse planes of the hip and knee joints for the jumping and landing phases; initial

contact (IC) joint angle values in the sagittal, frontal, and transverse planes of the hip and knee joint for the landing phase; execution time of the movement sequences in the jumping phase (jumping time); and the jump height.

## Statistical analysis

Statistical analysis was performed using SPSS (v23, IBM, Inc., Chicago, IL, USA) on the measurements of five trials of the SP and RT tasks. The paired samples t-test was used to determine differences in kinetics (maximal vGRF in jumping and landing phases), joint mechanics (joint angle and moments in jumping and landing phases), and performance measures (jumping time and jump height) during maximal single-leg CMJ between the SP and RT tasks. Cohen's d was also calculated to acquire the effect size for pairwise comparisons. Effect sizes for Cohen's d were classified as small (0.2), medium (0.5), and large (0.8) [36]. All statistical comparison was set at $p < 0.05$.

## Results

Representative joint angles (Figs 2 and 3) and moments (Figs 4 and 5) , time series of the hip (Figs 2 and 4) and knee (Figs 3 and 5) joints were selected to present as a graphical representation. The mean, standard deviation, *p*-value, and effect size (ES) for the peak joint angles and IC joint angles variables are presented in Table 1. During the jumping phase, significantly higher peak hip and knee flexion angles (p<0.001) were found in the SP task (Table 1). The participants performed single-leg CMJs by using 13 degrees less hip flexion and 9 degrees less knee flexion angles on average in the RT task. There were no peak joint angle differences in the frontal and transverse plane between the tasks. For the landing phase, hip ($p = 0.036$) and knee ($p = 0.038$) peak flexion angles were significantly higher in the SP task (Table 1).

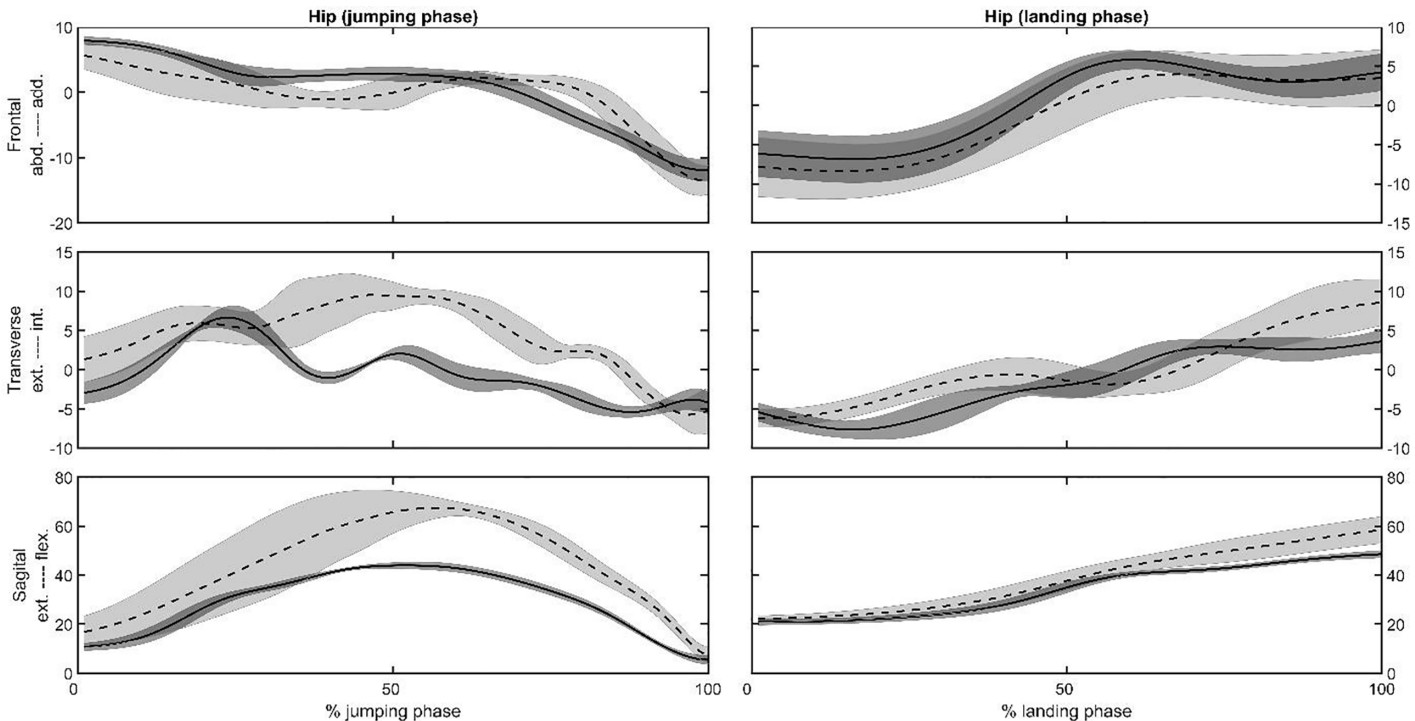

**Fig 2. Representative hip joint angles in the frontal, transverse and sagittal planes vs. the jumping or landing phases for a subject for the SP (dashed lines) and the RT (solid lines) tasks.** Shaded area is the plus and minus one between the subjects' standard deviation around the mean of the subject.

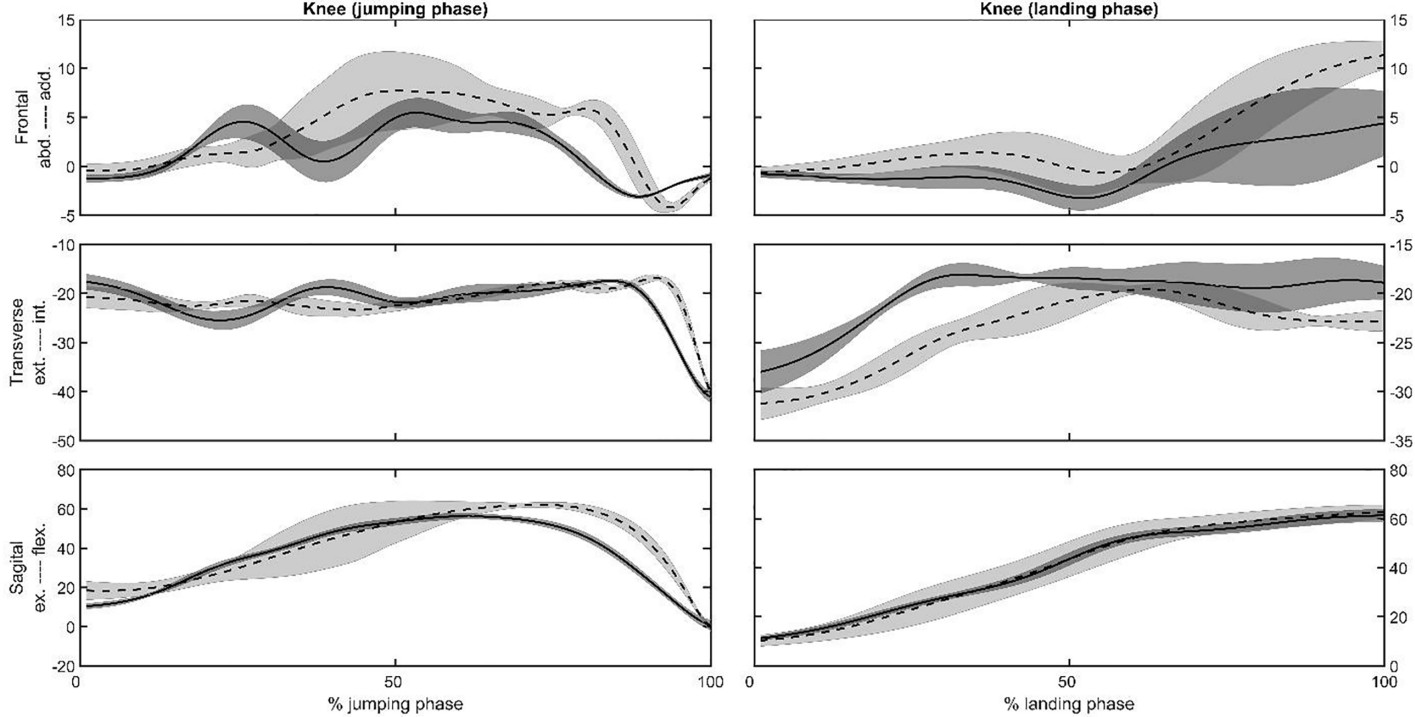

**Fig 3. Representative knee joint angles in the frontal, transverse and sagittal planes vs. the jumping or landing phases for the same subject for the SP (dashed lines) and the RT (solid lines) tasks.** Shaded area is the plus and minus one between the subjects' standard deviation around the mean of the subject.

The participants in the RT task used 5 degrees less hip and knee peak flexion angles on average in the landing phase. In addition to the peak angles, it was found that IC angles were similar in the sagittal, frontal, and transverse planes for SP and RT tasks ($p > 0.05$, Table 1).

The descriptive statistics (mean and standard deviation), $p$-value, and effect size of peak joint moments were presented in Table 2. In the frontal plane, greater hip ($p = 0.004$) and knee ($p = 0.006$) peak abduction moments were found in the jumping phase of the RT task, but there were no differences in the landing phase. When compared to the SP task, peak hip and knee abduction moments were 1.1 and 1.2 times greater in the jumping phase in the RT task respectively.

In addition to the frontal plane, in the sagittal plane, peak hip extension moment was greater in the SP task ($p = 0.006$) whereas peak knee extension moment was greater for the RT task ($p < 0.001$) in the jumping phase. The participants performed movement sequences in the jumping phase by using 1.3 times greater peak hip extension moment on average in the SP task and 1.2 times greater peak knee extension moment on average in the RT task. However, there were no significant differences in the sagittal plane joint kinetics in the landing phase between the tasks. Lastly, there were no significant differences in the transverse plane in terms of joint moments.

The mean, standard deviation, $p$-value, and effect size of the selected kinetic and performance variables were shown in Table 3. In the jumping phase, the peak vGRF was 1.2 times greater on average for the RT task ($p < 0.001$) (Table 3). However, there were no significant differences for peak vGRF between the tasks in the landing phase. As for performance variables, jumping time was significantly shorter (p < 0.001, 405 ms less in average) and jump height was significantly lower (2.3 cm on average, $p = 0.020$) in the RT task (Table 3).

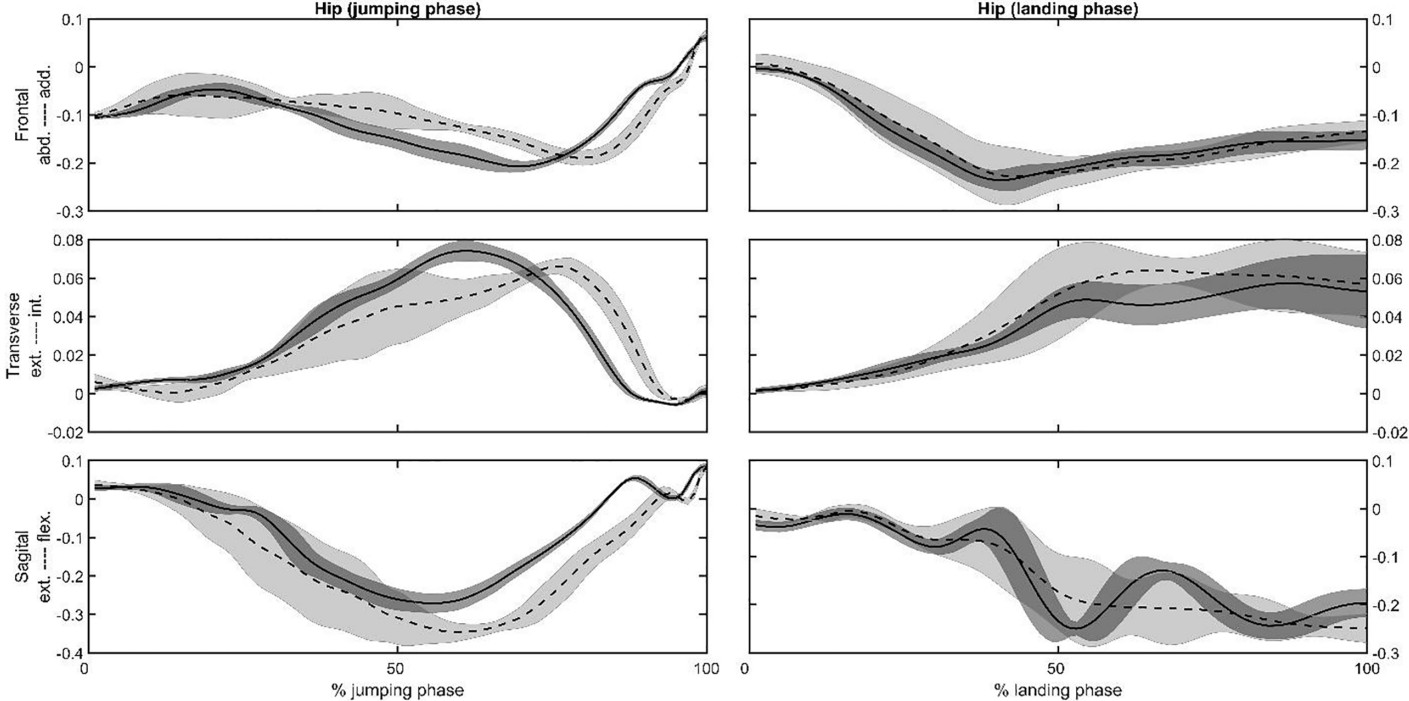

**Fig 4. Representative hip joint moments in the frontal, transverse and sagittal planes vs. the jumping or landing phases for the same subject for the SP (dashed lines) and the RT (solid lines) tasks.** Shaded area is the plus and minus one between the subjects' standard deviation around the mean of the subject.

## Discussion

The findings of this study partly supported our hypothesis that lower-limb kinetics and kinematics during single-leg maximal CMJ would differ under time pressure. Significant differences between SP and RT tasks were observed in the jumping phase for both kinematics and kinetics, while only kinematic differences emerged during the landing phase. From a motor control perspective, these findings align with McKinley and Pedotti (1992), who suggested that takeoff and landing phases are programmed independently [37].

Participants demonstrated reduced peak hip and knee flexion angles during both jumping and landing in the RT task. This reduction in range of motion (ROM) may have increased eccentric loading without improving jump height, consistent with prior studies [38,39]. Although RT trials showed higher peak vertical ground reaction forces (vGRF), jump height was lower, suggesting that time constraints may have limited effective energy transfer despite the increased loading. This supports the idea that reduced ROM and muscle slack reduction strategies (such as co-contractions and pretension) under time pressure may enhance force production but not necessarily improve jump performance [40,41]. Additionally, a possible the co-contraction strategy in the RT task may also invoke post-activation potentiation (PAP), a phenomenon where a preceding voluntary contraction enhances the performance of subsequent explosive movements. Etnyre and Kinugasa (2002) demonstrated that knee extension performed after three seconds a prior contraction resulted in a faster response [42]. This two possible mechanisms (i.e., pretension and PAP) may explain why, despite shorter flight time and reduced ROM in the RT task, participants were still able to achieve greater peak vGRF. Additionally, shorter flight times in the RT task may have restricted time for maximal force development in concentric phase. Salles et al. (2011) noted that reduced lower-limb muscle activation correlates with decreased joint flexion [43], which may explain the lower peak hip extension moments and shorter flight times observed under time pressure.

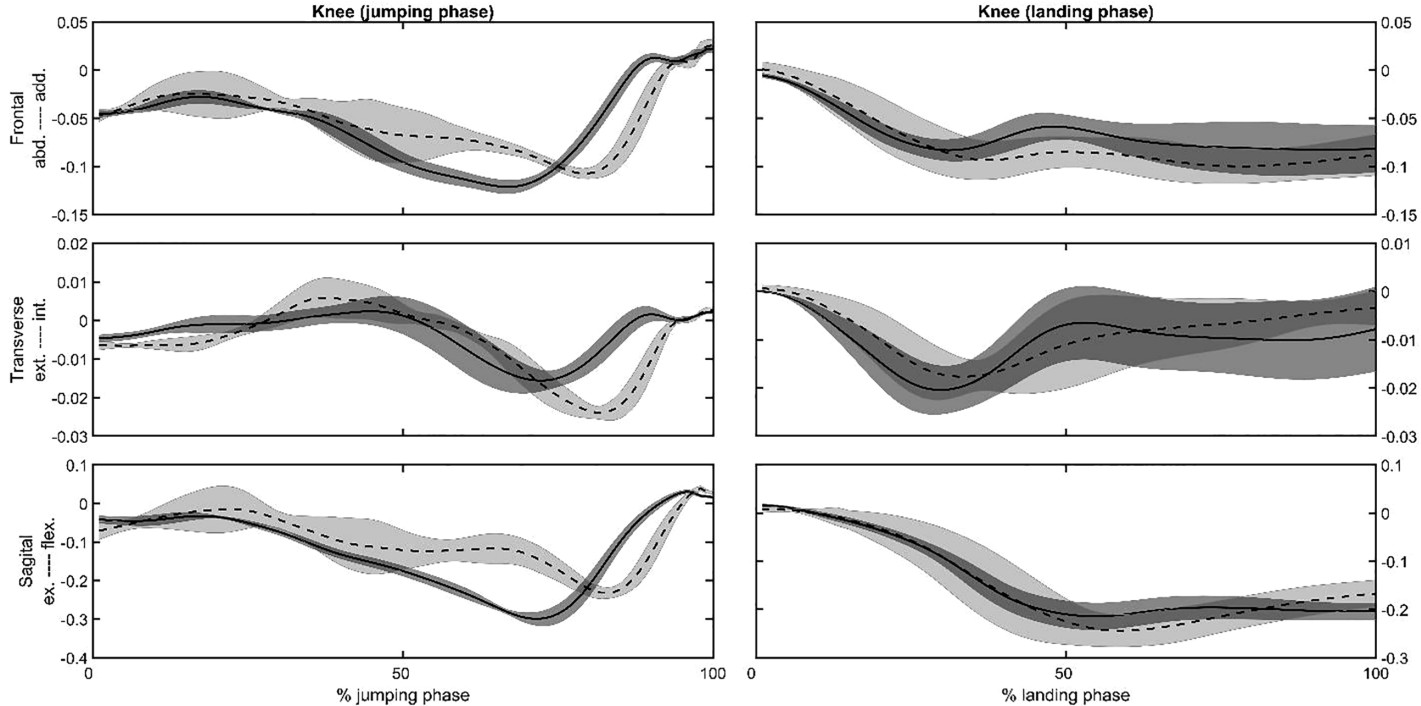

**Fig 5. Representative knee joint moments in the frontal, transverse and sagittal planes vs. the jumping or landing phases for the same subject for the SP (dashed lines) and the RT (solid lines) tasks.** Shaded area is the plus and minus one between the subjects' standard deviation around the mean of the subject.

Another key finding was the trade-off in joint strategies: participants showed greater hip extensor moments in the SP task and greater knee extensor moments in the RT task. This supports the idea that a "hip strategy" used in the SP task enabled longer flight time and higher jump height, while a "knee strategy" in the RT task allowed faster execution, albeit at the expense of optimal jump performance. Such adaptations are likely in game situations, where athletes prioritize speed due to time constraints.

In the landing phase, although initial contact angles, peak joint moments, and vGRF were similar between tasks, participants landed from higher jumps in the SP task and exhibited greater hip and knee flexion. This suggests a landing strategy involving more shock absorption and eccentric muscle activity, which likely helped reduce peak impact forces [44,45]. Conversely, reduced joint flexion at the end of landing in the RT task could increase joint loading, potentially elevating the risk of injuries such as ACL rupture, patellofemoral pain, or stress-related conditions [46,47]. Furthermore, unanticipated stimuli during the flight phase, common in dynamic sports, can alter landing mechanics and increase injury risk.

We also found greater hip and knee abduction moments in the RT task. While sagittal plane moments are central to jump propulsion [48,49], recent studies highlight the importance of frontal plane kinetics. Hip abductors not only stabilize the body but also contribute to propulsion via abduction torque [50–52]. However, excessive knee abduction moments are linked to increased ACL strain [53,54]. When combined with unfavorable foot placement (as may occur in competitive play), this could raise the risk of lower-limb injury [55].

While these findings provide valuable insight into biomechanical adaptations under time pressure, several limitations should be considered. First, although the RT task involved planned jumping under time constraints, actual sports environments include unexpected perturbations and external stimuli that may alter jump and landing mechanics mid-flight. Future studies should integrate unpredictable stimuli during the flight phase to better simulate game-like conditions. Second,

**Table 1. The mean, standard deviation (in mean±SD format), _p_-value, and effect size for the peak joint angles and IC joint angles in the frontal, transverse and sagittal planes for the RT and SP tasks in the jumping and/or landing phases.**

| Joint Angle | Phase | RT | SP | _p_ | ES |
|---|---|---|---|---|---|
| Hip add. | Jumping | 8.912±4.643 | 10.054±4.567 | .218 | 0.248 |
| | Landing | 5.318±3.917 | 5.321±4.567 | .997 | 0.001 |
| Hip int. rot. | Jumping | 7.279±6.013 | 7.301±4.694 | .984 | 0.004 |
| | Landing | 7.934±6.346 | 8.311±7.588 | .788 | 0.054 |
| Hip flex. | Jumping | 35.684±9.611 | 48.666±10.049 | **.000** | 1.32 |
| | Landing | 33.462±10.169 | 38.665±14.022 | **.036** | 0.425 |
| Knee add. | Jumping | 7.801±4.956 | 8.990±5.752 | .271 | 0.222 |
| | Landing | 8.567±4.901 | 11.492±9.494 | .056 | 0.387 |
| Knee ex. rot. | Jumping | −10.350±8.977 | −11.050±7.980 | .681 | 0.082 |
| | Landing | −8.645±9.940 | −10.085±8.704 | .443 | 0.154 |
| Knee flex. | Jumping | 50.155±8.637 | 59.786±7.133 | **.000** | 1.216 |
| | Landing | 48.441±8.671 | 53.231±13.596 | **.038** | 0.42 |
| Hip add. (IC) | Landing | −9.359±3.234 | −9.519±3.429 | .810 | 0.048 |
| Hip ex. rot. (IC) | Landing | −1.754±9.311 | −2.930±8.397 | .457 | 0.133 |
| Hip flex. (IC) | Landing | 17.386±6.077 | 17.944±7.184 | .676 | 0.084 |
| Knee add. (IC) | Landing | 4.140±3.188 | 3.660±3.327 | .464 | 0.147 |
| Knee ex. rot. (IC) | Landing | −28.392±9.320 | −27.773±9.555 | .744 | 0.066 |
| Knee flex. (IC) | Landing | 13.011±5.971 | 12.036±6.410 | .433 | 0.157 |

**Table 2. The mean, standard deviation (in mean±SD format), _p_-value, and effect size for the peak joint moments in the frontal, transverse and sagittal planes for the RT and SP tasks in the jumping and landing phases.**

| Joint Moment | Phase | RT | SP | _p_ | ES |
|---|---|---|---|---|---|
| Hip frontal | Jumping | −0.229±0.041 | −0.203±0.044 | **.004** | 0.611 |
| | Landing | −0.230±0.039 | −0.240±0.053 | .294 | 0.215 |
| Hip transverse | Jumping | 0.060±0.012 | 0.062±0.013 | .618 | 0.16 |
| | Landing | 0.059±0.016 | 0.061±0.015 | .500 | 0.129 |
| Hip sagittal | Jumping | −0.135±0.071 | −0.179±0.086 | **.006** | 0.558 |
| | Landing | −0.126±0.062 | −0,137±0.068 | .394 | 0.169 |
| Knee frontal | Jumping | −0.135±0.030 | −0.117±0.034 | **.006** | 0.561 |
| | Landing | −0.138±0.035 | −0.137±0.034 | .940 | 0.029 |
| Knee transverse | Jumping | −0.029±0.012 | −0.027±0.011 | .420 | 0.174 |
| | Landing | −0.025±0.014 | −0.026±0.015 | .851 | 0.069 |
| Knee sagittal | Jumping | −0.283±0.045 | −0.228±0.038 | **.000** | 1.321 |
| | Landing | −0.262±0.060 | −0.257±0.065 | .677 | 0.08 |

**Table 3. The mean, standard deviation (in mean±SD format), _p_-value, and effect size of the selected kinetic and performance variables for the RT and SP tasks.**

| Variable | RT | SP | _p_ | ES |
|---|---|---|---|---|
| Peak vGRF in jumping phase (BW) | 2.236±0.360 | 1.894±0.274 | **.000** | 1.07 |
| Peak vGRF in landing phase (BW) | 2.426±0.415 | 2.489±0.351 | .414 | 0.164 |
| Jump height (m) | 0.237±0.046 | 0.260±0.051 | **.020** | 0.474 |
| Jumping time (ms) | 517.2±137.1 | 922.8±286.1 | **.000** | 1.808 |

muscle activity was not recorded, limiting interpretation of preactivation or prestiffness strategies. Surface electromyography (EMG) would help clarify how muscle coordination patterns differ between time-constrained and unconstrained tasks. Lastly, although the number of participants was determined through power analysis and a paired-samples design was used to minimize inter individual variability, the sample size could still be considered a limitation. Future research should replicate these experiments with larger samples to test the generalizability of the findings.

## Conclusions

This study demonstrated that time pressure significantly alters lower-limb biomechanics during single-leg maximal CMJs, particularly in joint kinematics and kinetics during the jumping phase. While a "hip strategy" under self-paced conditions facilitated greater jump height and longer flight times, the "knee strategy" adopted under time constraints prioritized speed at the expense of performance. Despite higher peak vGRF in the reaction-time task, reduced ROM and shorter flight times suggest limited energy transfer efficiency, likely influenced by pretension and post-activation potentiation mechanisms. Altered frontal plane kinetics and decreased joint flexion during landing under time pressure may increase injury risk, particularly to the knee. These findings highlight critical performance-injury trade-offs in time-constrained scenarios and underscore the need for training interventions that enhance both explosive performance and neuromuscular control under game-like conditions. Future research incorporating unpredictable stimuli and EMG analysis is warranted to better understand muscle activation strategies and their implications for injury prevention.

## Author contributions

**Conceptualization:** Huseyin Celik, Pinar Arpinar-Avsar.

**Formal analysis:** Ugur Yilmaz, Huseyin Celik.

**Funding acquisition:** Pinar Arpinar-Avsar.

**Investigation:** Ugur Yilmaz.

**Software:** Huseyin Celik.

**Supervision:** Huseyin Celik, Pinar Arpinar-Avsar.

**Validation:** Huseyin Celik.

**Visualization:** Ugur Yilmaz.

**Writing – original draft:** Ugur Yilmaz.

**Writing – review & editing:** Huseyin Celik, Pinar Arpinar-Avsar.

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
