## [Decision Letter · Decision Letter 0]

4 Oct 2025

Dear Dr. Arpinar-Avsar,

Thank you for submitting your manuscript to PLOS ONE. After careful consideration, we feel that it has merit but does not fully meet PLOS ONE’s publication criteria as it currently stands. Therefore, we invite you to submit a revised version of the manuscript that addresses the points raised during the review process.

We look forward to receiving your revised manuscript.

Kind regards,

Emiliano Cè, Ph.D.

Academic Editor

PLOS ONE

Journal Requirements:

4. For studies involving third-party data, we encourage authors to share any data specific to their analyses that they can legally distribute. PLOS recognizes, however, that authors may be using third-party data they do not have the rights to share. When third-party data cannot be publicly shared, authors must provide all information necessary for interested researchers to apply to gain access to the data. (https://journals.plos.org/plosone/s/data-availability#loc-acceptable-data-access-restrictions)

Additional Editor Comments:

Dear Authors, one expert in the field reviewed your manuscript reporting several minor issues you should consider during the revision process.

Reviewer's Responses to Questions

**Comments to the Author**

1. Is the manuscript technically sound, and do the data support the conclusions?

Reviewer #1: Yes

2. Has the statistical analysis been performed appropriately and rigorously?

Reviewer #1: Yes

3. Have the authors made all data underlying the findings in their manuscript fully available?

Reviewer #1: Yes

4. Is the manuscript presented in an intelligible fashion and written in standard English?

Reviewer #1: Yes

Reviewer #1: The introduction provides an updated insight on the topic. Results are presented in an manner which is easy to follow.

One limitation which should be referred to is the limited number of volunteers taken into the study.

**Do you want your identity to be public for this peer review?** For information about this choice, including consent withdrawal, please see our Privacy Policy

Reviewer #1: **Yes: ** Diana Ciubotariu

---

## [Author Response · Author response to Decision Letter 1]

9 Oct 2025

We sincerely thank the Reviewer for these encouraging comments. In response to the following statement from the Reviewer in the ‘5. Review Comments to the Author’ section:

Reviewer #1: The introduction provides an updated insight on the topic. Results are presented in an manner which is easy to follow. One limitation which should be referred to is the limited number of volunteers taken into the study.

We have added the following sentence to the Limitations paragraph, which now reads (Line 316-319):

"Although the number of volunteers included in the study was based on power analysis and we selected a paired samples design to reduce variability due to individual differences between subjects, the sample size could be considered a limitation of the study. Future studies could replicate these experiments with larger sample sizes to test the generalizability of our findings."

Additional corrections:

• Funding Statement: We have removed the funding paragraph from the main manuscript. The funding statement is: “This research was partially supported by TUBITAK under Grant 115S535.” was added to the relevant section in editorial manager.

• Author order: The author order has been correct in the originally submitted manuscript since the first version; however, it was entered incorrectly in Editorial Manager due to confusion about the up/down reordering controls. We have now updated Editorial Manager to match the author order submitted in the manuscript. We did not add/remove/replace aouthors.

• Data sharing: We have added the following statement to the manuscript: “All data generated or analyzed during this study are available in the Zenodo repository at https://doi.org/10.5281/zenodo.17284783 .”

---

## [Decision Letter · Decision Letter 1]

6 Nov 2025

Time Pressure Alters Take-Off But Not Landing Biomechanics In Single-Leg Countermovement Jumps

PONE-D-25-38177R1

Dear Dr. Arpinar Avsar,

We’re pleased to inform you that your manuscript has been judged scientifically suitable for publication and will be formally accepted for publication once it meets all outstanding technical requirements.

Kind regards,

Emiliano Cè, Ph.D.

Academic Editor

PLOS ONE

Additional Editor Comments (optional):

Reviewers' comments:

Reviewer's Responses to Questions

**Comments to the Author**

Reviewer #1: All comments have been addressed

2. Is the manuscript technically sound, and do the data support the conclusions?

Reviewer #1: Yes

3. Has the statistical analysis been performed appropriately and rigorously?

Reviewer #1: Yes

4. Have the authors made all data underlying the findings in their manuscript fully available?

Reviewer #1: Yes

5. Is the manuscript presented in an intelligible fashion and written in standard English?

Reviewer #1: (No Response)

Reviewer #1: I consider that the paper is well designed and that the Discussion chapter properly integrates the finding within medical literature.

**Do you want your identity to be public for this peer review?** For information about this choice, including consent withdrawal, please see our Privacy Policy

Reviewer #1: **Yes: ** Diana Ciubotariu

---

## [Editor Report · Acceptance letter]

PONE-D-25-38177R1

PLOS ONE

Dear Dr. Arpinar-Avsar,

I'm pleased to inform you that your manuscript has been deemed suitable for publication in PLOS ONE. Congratulations! Your manuscript is now being handed over to our production team.

Kind regards,

on behalf of

Prof. Emiliano Cè

Academic Editor

PLOS ONE